# Cross-Domain Transfer Learning Prediction of COVID-19 Popular Topics Based on Knowledge Graph

**Xiaolin Chen** [1] , **Qixing Qu** [2,*] , **Chengxi Wei** [3] **and Shudong Chen** [1,4]

1 Institute of Microelectronics of the Chinese Academy of Sciences, Beijing 100029, China; chenxiaolin@ime.ac.cn (X.C.); chenshudong@ime.ac.cn (S.C.)
2 School of Information, University of International Business and Economics, No.10 Huixin East Street, Chaoyang District, Beijing 100029, China
3 Xiangsihu College of Guangxi University for Nationalities, Nanning 530031, China; cxwei0815@gmail.com
4 School of Microelectronics, University of Chinese Academy of Sciences, Beijing 100049, China
* Correspondence: qqxing@uibe.edu.cn

**Abstract:** The significance of research on public opinion monitoring of social network emergencies is becoming increasingly important. As a platform for users to communicate and share information online, social networks are often the source of public opinion about emergencies. Considering the relevance and transmissibility of the same event in different social networks, this paper takes the COVID-19 outbreak as the background and selects the platforms Weibo and TikTok as the research objects. In this paper, first, we use the transfer learning model to apply the knowledge obtained in the source domain of Weibo to the target domain of TikTok. From the perspective of text information, we propose an improved TC-LDA model to measure the similarity between the two domains, including temporal similarity and conceptual similarity, which effectively improves the learning effect of instance transfer and makes up for the problem of insufficient sample data in the target domain. Then, based on the results of transfer learning, we use the improved single-pass incremental clustering algorithm to discover and filter popular topics in streaming data of social networks. Finally, we build a topic knowledge graph using the Neo4j graph database and conduct experiments to predict the evolution of popular topics in new emergencies. Our research results can provide a reference for public opinion monitoring and early warning of emergencies in government departments.

**Keywords:** transfer learning; cross-domain prediction; COVID-19; popular topics; knowledge graph

## 1. Introduction

Social networks have changed the form of information dissemination about public emergencies from "limited reporting" to "nationwide communication". User-generated content (UGC) has emerged in large numbers, making every citizen both the receiver and disseminator of information. In 2020, the outbreak of COVID-19 triggered the biggest public panic in human history. As the first platform to form and disseminate information related to emergencies, social networks have become the incubator and catalyst for mass panic, which has played a driving role in promoting information dissemination. However, this pattern makes a lot of information interweave, which leads to the problem of information overload. As human attention to information is limited, attention becomes a scarce resource [1]. In the process of information transmission, the distribution of popularity is uneven; most of the information has low popularity, and only a small portion of the information can maintain high popularity. In this case, what information will attract people's attention and how it changes over time become the focus of research [2]. To solve this problem, it is necessary to make full use of network text data and construct a scientific and effective prediction model to analyze the potential evolution of popular topics.

While most previous studies only focused on information in a single network, social networks are interconnected, and the same event spread in different social networks has

relevance and transitivity. For example, when an emergency breaks out on Weibo, it will drive the clicks of its related content on video websites, so cross-domain information association is needed. Based on this, this paper has two research questions:

(1) How can one measure the text information similarity of different social network platforms for knowledge transfer learning?
(2) From the perspective of text information, how can one detect potential popular topics on social networks and predict popular topics?

This paper adopts the transfer learning algorithm to predict the popular topics of emergencies in combination with information from other social network platforms to improve the accuracy of prediction. Based on the transfer learning results, this paper combines the knowledge graph with the emergency data to construct the public opinion topic graph, which will present the network structure of social network topics with semantic characteristics and then reach the goal of tracking the evolution of popular topics. The basic research framework of this paper is shown in Figure 1. The research results help government departments to guide practice, grasp the public opinion trends of emergencies, and realize monitoring analysis and early warning of public opinion so as to take scientific and effective measures to guide and control public opinion.

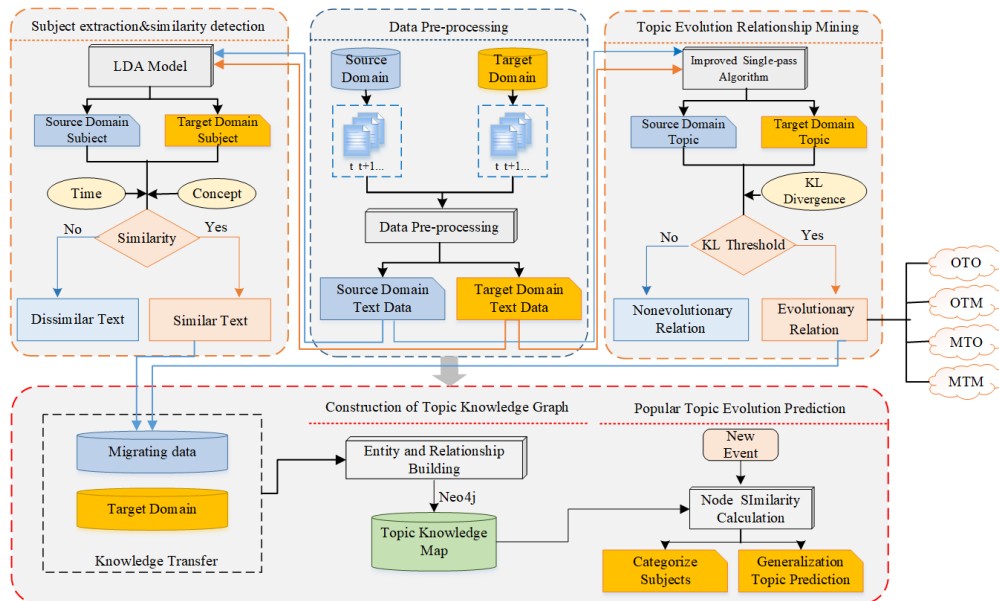

**Figure 1.** The basic research framework of the study. Based on the preprocessed text data: (1) For data of the same time slice, the LDA model was used to extract similar subject text data for transfer learning; (2) for data of different time slices, the single-pass algorithm was used to extract potential topics and evolutionary relationships; (3) the above results are used to construct the topic knowledge graph and predict popular topics.

## 2. Related Work

At present, for research on topic detection, scholars mostly work on methods such as topic modeling, multidimensional attribute extraction and algorithm-based improvement perspectives to identify the Internet users' topics [3–5]. The majority of them adopt multiple perspectives and techniques to detect topics in online public opinion events, but little work on the field of public opinion has been done with the help of knowledge graph methods [6–9]. The topic graph constructed by combining knowledge graph with data of online public opinion emergencies will present a complete network structure reflecting comment forwarding relationships and with semantic characteristics. Through the extraction of multi-dimensional feature attributes in the graph, the objective of tracking the evolution of topics in a comprehensive and systematic manner is achieved. We discuss related work in the fields of transfer learning, popular topic detection, and knowledge graphs.

Transfer learning. This concept first originated in psychology, arguing that people can use previously learned knowledge and skills to guide the learning of new knowledge [10]. Common machine learning techniques traditionally address isolated tasks. In contrast, transfer learning aims to transfer knowledge learned in one source domain and use it to improve learning in a related target domain [11]. Transfer learning has been previously used in various cases including classification, image clustering, collaborative filtering, and sensor-based location prediction [12–14]. The feature learning problem of little or no sample data in the target domain can be solved by transferring the source domain data [15].

Popular Topic Detection. Researchers have investigated a wide variety of methods and resources to detect popular topics of online information [16], e.g., the LDA model replaces the traditional VSM model to extract hidden topic information [17], a method of clustering words based on the similarity of related time series [18], overview-based topic models and real-time detection techniques [19], incremental clustering to detect new topics and use content and temporal features to discover popular topics in time [20], and layered topic detection methods [21].

Knowledge Graph. The essence of a knowledge graph is a semantic network graph that reveals entity knowledge [22]. Scholars use the vocabulary provided by the semantic network to achieve short text understanding, word segmentation, type annotation, and concept labeling [3]. Knowledge graph technology has made remarkable achievements in medicine, film and television, transportation, and other fields, but there is still a lot of work to be done in the field of social networking [23,24].

## 3. Data Pre-Processing

This paper takes the novel coronavirus pneumonia in 2020 as its research scenario. We collected datasets from the Harvard Dataverse platform [25]. The two main social network platforms of Weibo and TikTok were taken as the research object. Based on transfer learning, the cross-domain prediction problem of the prevalence of new text messages on the social network platform was studied.

Weibo COVID dataset. Weibo, commonly referred to as "Chinese Twitter", is a micro-blogging site. The data was crawled on the Weibo platform from 7 December 2019 to 4 April 2020. The data was crawled in two phases covering a total of 4,047,389 Weibo posts. The first crawler ran on 26 February 2020 and collected 3.3 million Weibo posts from 18 January 2020 to 26 February 2020. The second crawler ran on 4 April crawling from 7 December 2020 to 4 April 2020 to complement the original dataset.

TikTok COVID dataset. We mainly used the GooSeeker big data crawler software to crawl the data of the TikTok website. Data were collected from 25 December 2019 to 30 May 2020, crawling a total of 15,756 pieces of data from 2685 government media users. Since the short videos were released by government media accounts, they have high authority, a wide spread, and great influence.

We adopted Chinese word segmentation (jieba) to segment Chinese text, and added a user-defined dictionary (dict. Txt) based on the new crown open concept knowledge graph publicly released on the OpenKG platform (COVID-19-Concept) to optimize the word segmentation results. Additionally, this study makes comprehensive use of the advantages of "Baidu deactivation Thesaurus", "Harbin Institute of technology deactivation Thesaurus" and "Sichuan University Machine Intelligence Laboratory deactivation Thesaurus" to filter stop words [26,27]. We combined the contents of the three deactivation thesauruses to build a new deactivation thesaurus and deleted meaningless information so as to reduce the interference to the word segmentation results.

After the above processing, the Chinese text data was still in text format and could not be directly recognized and calculated by the computer. Therefore, text representation processing is required. This paper represents text based on the vector space model (VSM). Then characteristic words with strong discrimination and representativeness were extracted from the text so as to reduce the dimension of vector space, simplify the calculation process, and improve the efficiency of text processing without damaging the core information. At

the same time, the word frequency was normalized to avoid the interference of the length of the text. In this paper, term frequency inverse document frequency (TF-IDF) was used to extract text features. If a word had a higher word frequency in one document and a lower word frequency in other documents, it was considered that the word could distinguish documents well and it was given a greater weight; On the contrary, if the word appears in multiple documents, it indicates that its distinguishing ability is not strong, and the value of IDF is small. The whole process of data preprocessing is shown in Figure 2. Finally, we used the obtained data for model construction and analysis.

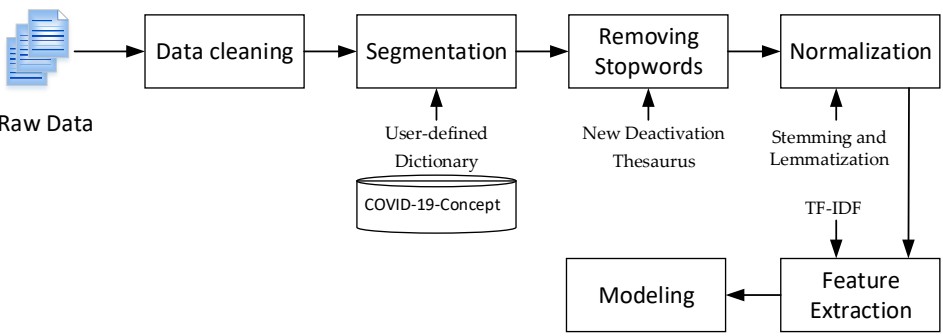

**Figure 2.** The process of text data preprocessing and optimization methods.

## 4. Transfer Learning Model

In this paper, we adopted the transfer learning model which was established based on the DSN Network [28]. This method considers that the source domain $D_s$ and the target domain $D_t$ are composed of two parts: the public part and the private part. The former can learn the public features and the latter is used to keep the independent characteristics of each domain. Traditional machine learning algorithms require the training and testing sample data to have the same distribution. However, the transfer learning algorithm broke the limitations, as long as there is a correlation between the $D_s$ and $D_t$ domains, knowledge can be obtained from the $D_s$ domain which can be applied to the different but similar $D_t$ domain to solve the problem of insufficient $D_t$ training samples. Therefore, this paper establishes a transfer learning model based on the DSN network method.

According to the principle of similarity matching, instance data with high similarity between the source domain $D_s$ dataset (Weibo platform) and the target domain $D_t$ dataset (TikTok platform) were transferred to the target domain to help with target domain model learning. In the transfer learning model, the source domain was set as $D_s$ and the learning task in the source domain was denoted as $T_s$ and the target domain was set as $D_t$ and the learning task in the target domain was denoted as $T_t$. The data distribution of these two domains is $P(X^s)$ and $P(X^t)$, and $P(X^s) \neq P(X^t)$. In order to improve the performance of transfer learning, the $D_t$ domain data was divided into training sample set $X = \{X_1, X_2, \cdots, X_u\}$ and test sample set $Y = \{Y_1, Y_2, \cdots, Y_v\}$, (i.e., $D_t = X \cup Y$), where $u$ and $v$ represent the time sequence length, respectively. Then, $D_s$ and $X$ were combined into the training set. The structure of the DSN network transfer learning algorithm is shown in Figure 3.

In the Figure 3, $h_c^t$ and $h_c^s$ are the hidden vectors of common features extracted from the $D_s$ and $D_t$ domain. $h_p^t$ and $h_p^s$ are the hidden vectors of private features extracted from the $D_s$ and $D_t$ domain. $F_{common}$ is the similar feature extracted from the $D_s$ and $D_t$ domain by calculating the similarity between $h_c^s$ and $h_c^t$, and $F_{difference}$ is the dissimilar feature extracted from $D_s$ and $D_t$ domain, which ensures that the private part still plays a role in learning task $T_s$ and $T_t$. $F_{latent}$ is the latent feature which is further mined based on dissimilar features. $\varphi_{similarity}$ is an indicator to measure the similarity between platforms, which is calculated by two dimensions: temporal similarity and conceptual similarity. Based on the $\varphi_{similarity}$ value, the paper transfers the source domain samples to the target domain for training research.

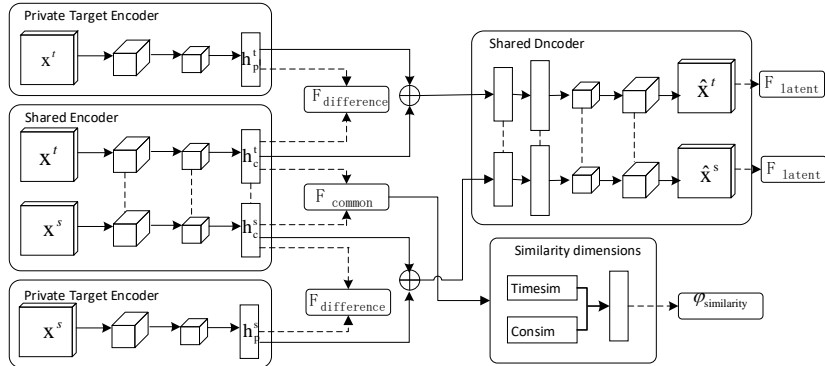

**Figure 3.** Structure diagram of transfer learning algorithm based on DSN Network.

### 4.1. The Improved TC-LDA Model

The LDA (Latent Dirichlet Allocation) model is a classic subject model. This paper uses the distribution of words in the text to find the potential subject by clustering words with similar distribution (Figure 4). After LDA model subject extraction, the text set of the source domain is represented as $D_s = \{v_1, v_2, \cdots v_k, p_j\}$, which is composed of feature word $\{v_1, v_2, \cdots v_n\}$ and subject word $p_j$. Similarly, the text set extracted from the training set of the target domain is represented as $X = \{f_1, f_2, \cdots f_m, d_i\}$, consisting of feature word $\{f_1, f_2, \cdots f_u\}$ and subject word $d_i$.

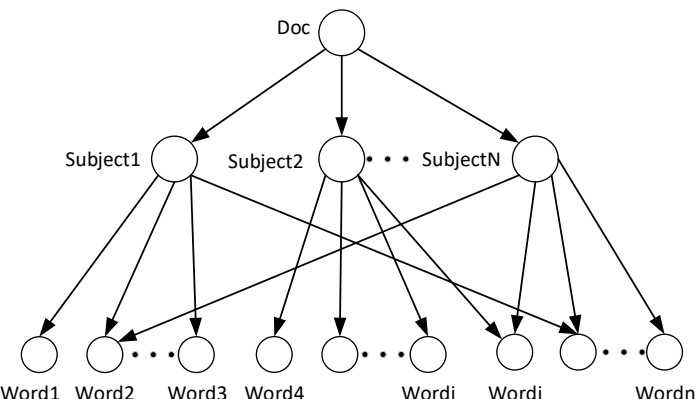

**Figure 4.** LDA model potential subject topology diagram.

As the source domain and target domain of this study are both social network media, the text content belongs to the short text type, which has the characteristics of time dynamic and feature sparsity of short text. In view of the characteristics of short texts, this paper improves the LDA model to calculate text similarity and proposes TC-LDA similarity model. First, considering the influence of the time factor, time feature similarity is introduced. Since this paper studies hot topics on social media, the popularity will drop significantly after 3 days. Therefore, the time interval is set as 3 days (72 h) in this paper. The longer the time interval, the lower the similarity value of the time feature between texts with more than 3 days. We take text $w_1$ in the source domain and text $w_2$ in the target domain for example. For texts with the same time, a greater weight is assigned, and the calculation is shown in Formula (1):

$$\text{Timesim}(w_1, w_2) = \begin{cases} 100 & w_2.Time = w_1.Time \\ \frac{72}{|w_2.Time - w_1.Time|} & w_2.Time \neq w_1.Time \end{cases} \tag{1}$$

Second, for the feature sparsity of short texts, concept similarity is introduced to expand the feature space of short texts.

The calculation formula of the number of concept words shared by two texts $c_i$ is as follows:

$$\text{N}(c_i|w_1 \cap w_2) = \text{Conof}(w_1) \cap \text{Conof}(w_2) \tag{2}$$

The formula for calculating the total number of concept words contained in the two texts $c_j$ is as follows:

$$\text{N}(c_j|w_1 \cup w_2) = \text{Conof}(w_1) \cup \text{Conof}(w_2) \tag{3}$$

The calculation formula of concept similarity is:

$$\text{Consim}(w_1, w_2) = \frac{\text{N}(c_i|w_1 \cap w_2)}{\text{N}(c_j|w_1 \cup w_2)} \tag{4}$$

The definition of the similarity calculation formula in this paper is given. For text $w_1$ in the source domain and text $w_2$ in the target domain, under the condition that they are divided into the same subject class through the analysis of LDA subject model, through the feature weight coefficient λ, combined with the two factors of time feature similarity Timesim$(w_1, w_2)$ and concept similarity Consim$(w_1, w_2)$, $\beta_{w1}$ and $\beta_{w2}$ are the subject weights of text division obtained by LDA model. After testing, the similarity threshold of 0.3 is found to be the most appropriate. The calculation of determining the similarity between $w_1$ and $w_2$ is shown in Formula (5):

$$\text{Similarity}(w_1, w_2) = (\lambda * \text{Timesim}(w_1, w_2) + (1 - \lambda)\text{Consim}(w_1, w_2)) * \beta_{w1} * \beta_{w2} \tag{5}$$

### 4.2. Experiment

Based on previous research results and combined with the characteristics of COVID-19 emergencies [29], this paper divides the development of epidemic public opinion into six stages: incubation period, growth period, outbreak period, fluctuation period, decline period, and long tail period, as is depicted in Figure 5. Additionally, in order to more accurately reflect the topic content in each period, according to the statistical analysis results of the life cycle stages, this paper divided data into time slices in the granularity of one week (7 days). The first 70% of the data in the two domains was selected as the training set, including 14 time slices (T0, T1, . . . , T14) from 25 December 2019 to 7 April 2020). The last 30% of the data was selected as the testing set, including 7 time slices (T15, T16, . . . , T21) from 8 April 2020 to 31 May 2020. The data were then allocated to the corresponding time slice, and the text in each time slice was processed in turn and the subject extracted.

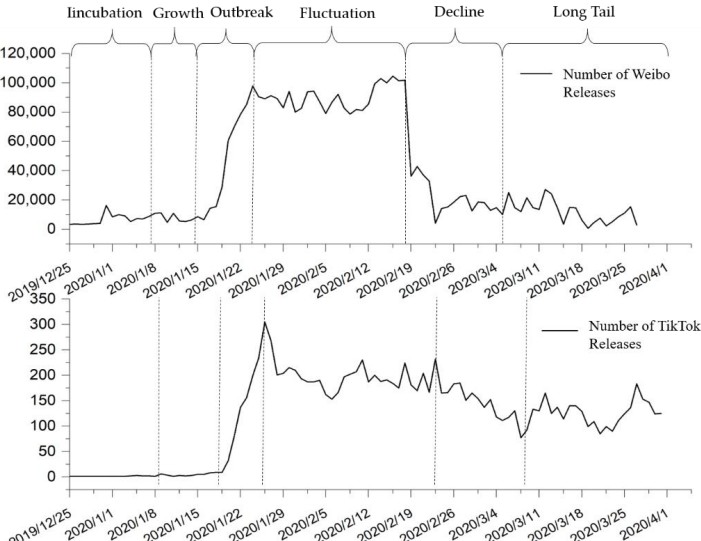

**Figure 5.** Stage division of life cycle of Weibo and TikTok platform events.

The data used in this experiment are the data from the Weibo and TikTok platforms after data pretreatment. This paper takes the LDA model processing of source domain and target domain data on T1 time slice as an example to illustrate. Data on other time slices were processed according to the similar LDA model mentioned above.

Through Gensim's LDA Model method, the subject number range was set as [2,20], and the subject consistency scores of LDA models in the source domain and target domain under different subject numbers were obtained, as shown in Figure 6. The best number of subjects for the source domain T1 time slice data was obtained as 16, and the model's perplexity is −6.9. For subjects in the target domain, the optimal number was 12, with the model's perplexity equal to −5.27.

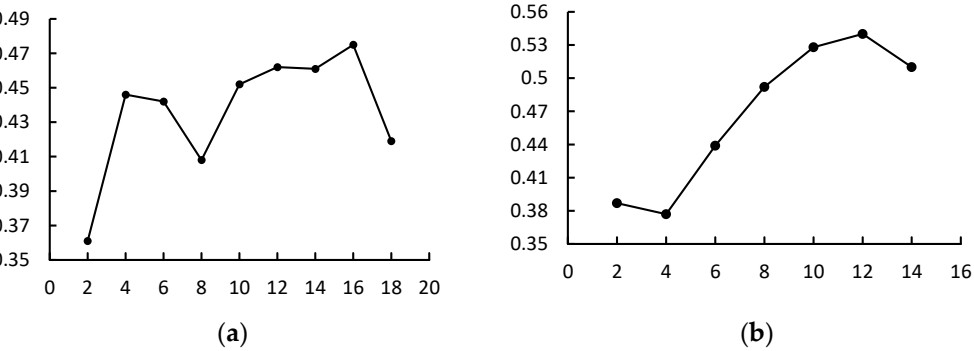

(**a**)                    (**b**)

**Figure 6.** This is the subject consistency of T1 time slice data in the two domains: (**a**) is the subject consistency score of source domain T1 time slice data; (**b**) is the subject consistency score of target domain T1 time slice data.

Then we set the number of subjects in the source domain to 16, the number of subjects in the target domain to 12, and reserved 10 keywords for each subject. Check the subject results extracted from the source domain and target domain in Figure 7.

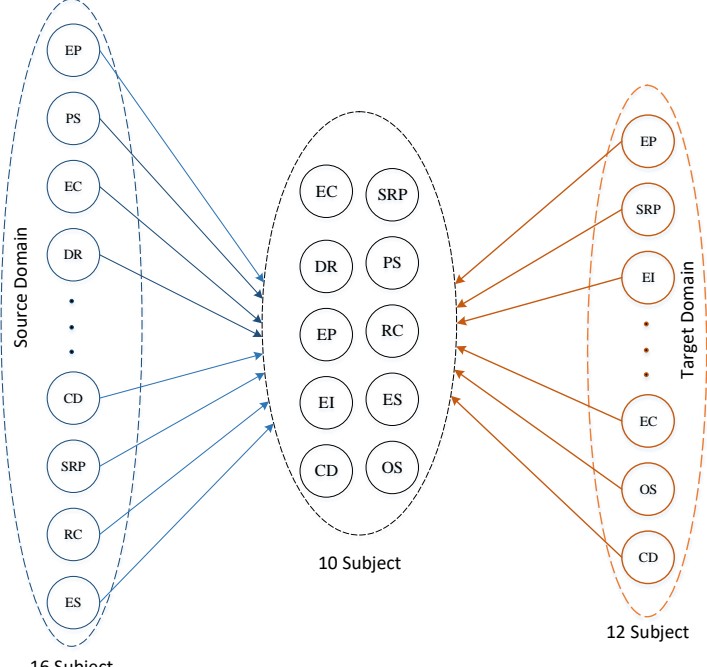

**Figure 7.** This is the subject classification of the source domain and target domain T1 time slice data: The T1 time slice of the source domain has 16 subjects; The T1 time slice of the target domain has 12 subjects. There are 10 subjects in the union of the two domains.

Each subject is composed of multiple words, and the same word may be contained in multiple subjects. For example, the 10 key words of the EP subject include "prevention and control, COVID-19, epidemic, combat, Novel Coronavirus, pneumonia, gather, disinfection, and Wuhan" (note: original data keywords are Chinese; this is translated data). Based on the above analysis, this paper is mainly divided into 10 subjects: epidemic condition (EC), data reporting (DR), epidemic prevention (EP), epidemic impact (EI), charity donations (CD), scientific research and popularization (SRP), people's stories (PS), rumor clarification (RC), emotional subjects (ES), and other subjects (OS).

### 4.3. Results

According to the results of subject extraction from LDA models of the source domain and target domain, we proposed an improved TC-LDA model to calculate the similarity of texts with the same subject. Some text data are extracted for explanation, as shown in Table 1.

**Table 1.** Examples of the texts of the same topic in $D_s$ and $D_t$ (T1 time slice).

| Num | Domain | Release Time | Subject | Weight | Content |
|-----|--------|--------------|---------|--------|---------|
| 1 | Ds | 2 January 2020 22:29 | Subject 16 | 0.7808 | SRP |
| | Dt | 5 January 2020 23:18 | Subject 12 | 0.6944 | SRP |
| 2 | Ds | 3 January 2020 19:13 | Subject 13 | 0.8815 | EP |
| | Dt | 6 January 2020 9:03 | Subject 2 | 0.9167 | EP |
| 3 | Ds | 6 January 2020 14:27 | Subject 4 | 0.4109 | ES |
| | Dt | 1 January 2020 8:00 | Subject 5 | 0.5085 | ES |

According to Formula (1) given by TC-LDA, the temporal similarity results of the above three groups of text examples are calculated (see Table 2). For conceptual similarity, we used the open API interface of the CN-Probase concept knowledge graph platform to query the concept words of related entity words in the text. This approach can expand feature words and reduce feature sparsity in short texts. Then, we used Formula (4) to calculate the conceptual similarity of text feature words, and the results are shown in Table 3. Finally, the similarity between the two text message fields was obtained according to Formula (5). For information with high similarity, instance transfer is performed, and information with low similarity is retained to keep the private features of source and target domains.

**Table 2.** Temporal similarity of T1 time slice data in the $D_s$ and $D_t$.

| Num | Domain | Release Time | Time Interval (Hours) | Temporal Similarity |
|-----|--------|--------------|----------------------|---------------------|
| 1 | Ds | 2 January 2020 22:29 | 72.82 | 0.99 |
| | Dt | 5 January 2020 23:18 | | |
| 2 | Ds | 3 January 2020 19:13 | 61.83 | 1.16 |
| | Dt | 6 January 2020 9:03 | | |
| 3 | Ds | 6 January 2020 14:27 | 126.45 | 0.57 |
| | Dt | 8 January 2020 8:00 | | |

In order to measure the effect of text similarity calculation, we adopted *Accuracy* and *F − Measure* values to measure the effect of text similarity. Parameters related to *Accuracy* and *F − Measure* are shown in Table 4.

**Table 3.** Conceptual similarity of T1 time slice data in the $D_s$ and $D_t$.

| Num | Domain | Shared Words | The Total Number of Words | Conceptual Similarity |
|:---:|:---:|:---:|:---:|:---:|
| 1 | Ds<br>Dt | 8 | 16 | 0.5 |
| 2 | Ds<br>Dt | 6 | 16 | 0.375 |
| 3 | Ds<br>Dt | 2 | 28 | 0.071 |

**Table 4.** Accuracy and F-measure Parameters.

| Num | Classification | Similar | Dissimilar |
|:---:|:---:|:---:|:---:|
| 1 | True | TP | TN |
| 2 | False | FP | FN |

*Accuracy* refers to the ratio of correctly classified text data to all texts, which is used to evaluate the performance of correct migration of similar texts in source domain and target domain. $F - Measure$ is used to evaluate the overall performance of similar text transferring in the source domain and target domain. It is calculated by *Procision* and *Recall*. The formula is defined as follows:

$$Accuracy = \frac{TP + TN}{TP + TN + FP + FN} \tag{6}$$

$$Procision = \frac{TP}{TP + FP} \tag{7}$$

$$Recall = \frac{TN}{TN + FN} \tag{8}$$

$$F - Measure = 2 \times \left( \frac{Procision \times Recall}{Procision + Recall} \right) \tag{9}$$

In order to obtain better experimental results, this paper sets the characteristic weight coefficient to change from 0.1 to 1 and carries out experiments according to the value of different parameters. According to the principle of TC-LDA model in Section 4.1, if the similarity threshold is set to 0.3, that is, for texts with a Similarity$(w_1, w_2) \geq 0.3$, it is considered that the texts of the source domain and the target domain are similar, and transfer learning can be carried out. Otherwise, the texts are considered to be dissimilar samples, and the results are shown in Figure 8. When the feature weight coefficient $\lambda = 0.5$, the *Accuracy* and $F - Measure$ value of TC-LDA model are the highest, so we set $\lambda = 0.5$ in this study.

In order to measure the performance of the improved TC-LDA model, this paper compares the LDA model with the TC-LDA model through four evaluation criteria, and the results are shown in Table 5. We can see that the TC-LDA model proposed in this study has significant advantages over the LDA model in similar text detection. The new model improves the *Accuracy*, *Procision*, *Recall* and $F - Measure$ value, which contributes to the subsequent transfer learning and data mining.

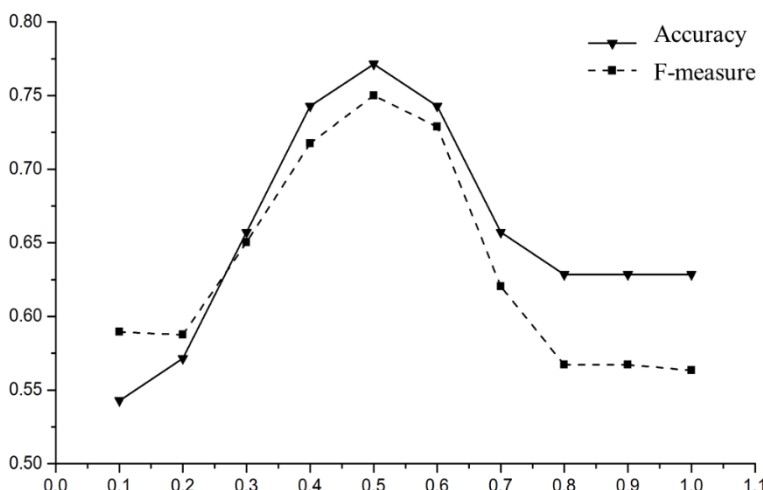

**Figure 8.** *Accuracy* and *F − Measure* change in feature weight coefficient λ = [0.1, 1].

**Table 5.** Performance comparison between LDA model and TC-LDA model.

| Model | Accuracy | Precision | Recall | F-Measure |
|-------|----------|-----------|--------|-----------|
| LDA | 57.14% | 76.92% | 45.45% | 57.14% |
| TC-LDA | 77.14% | 81.82% | 69.23% | 75.00% |

## 5. Prediction of Popular Topics Based on Knowledge Graph

According to the results of transfer learning in Section 4, for text data with low similarity in time slices, transfer learning cannot compensate for training samples in the target domain. Therefore, the source domain and target domain retain their private features respectively, but text topics in different time slices may also have an evolutionary relationship with relevance. Therefore, the potential topic information can be extracted from the source domain which can be transferred to the target domain in the adjacent time slice for the prediction of potential popular topics in the target domain.

### 5.1. Topic Discovery and Filtering

Subject extraction of text data of each time slice belongs to subject discovery of static text information. However, social network texts are particularly affected by time, so this study adopts single-pass algorithm to find topics [30]. Due to the large amount of data, this study takes the data of the T2 time slice in the source domain and the T3 time slice in the target domain as an example to illustrate the topic discovery process of single-pass algorithm. The calculation steps for data in the other time slices are the same.

Aiming at the shortcomings of the single-pass algorithm [31], we made the following improvements to this algorithm: On the one hand, we sorted text data according to the release time before input, which was suitable for the evolution of the topic. In this way, we reduced the influence of different input sequences on clustering results. On the other hand, we use the way of the inverted index to accelerate the clustering and improve the search efficiency when the data volume is large.

After several experiments, it was found that it is appropriate to define the threshold as 0.2 and realize the model through Python software to obtain the respective topic categories of the source domain and target domain. As machine learning cannot fully understand human natural language text, limitations exist in the topic filtering. Therefore, we manually set "problem-answer" to train the binary classifier, which is used to filter topics after training to some texts. If relevant, they will be left, while irrelevant topics will be deleted. In this way, we filter out the topics unrelated to COVID-19 events. The final number of topic categories is shown in Table 6.

**Table 6.** The number of topics extracted by single-pass in T2 and T3.

| Domain | Time | Number of Topics Extracted by Single-Pass | Number of Filtered Topics |
|---|---|---|---|
| Ds | 8 January 2020 | 91 | 5 |
| | 9 January 2020 | 61 | 6 |
| | 10 January 2020 | 23 | 5 |
| | 11 January 2020 | 17 | 5 |
| | 12 January 2020 | 37 | 4 |
| | 13 January 2020 | 99 | 5 |
| | 14 January 2020 | 39 | 7 |
| Dt | 15 January 2020–21 January 2020 | 25 | 17 |

In this section, there were a total of nine subjects involved in the source domain and target domain data, which are encoded by letters [a–i], respectively. As for the extraction results, the topics were manually labeled according to topic category keywords and topic sentences and were encoded by 1 to 10, respectively. The specific results are shown in Table 7.

**Table 7.** Topic extraction results extracted by single-pass in T2 and T3.

| Domain | Num | EP(a) | DR(b) | SRP(c) | CD(d) | EI(e) | EC(f) | OS(g) |
|---|---|---|---|---|---|---|---|---|
| Ds-T2 Topics | 1 | Mask purchase Publicity | Eight patients with viral pneumonia of unknown cause have been discharged from hospital in Wuhan | SARS has been ruled out as the cause of pneumonia of unknown cause in Wuhan | Face mask shortage | Entertainment industry field influence | Us flu outbreak | Australian bushfire |
| | 2 | Wildlife quarantine | People are demanding transparency | The pathogen of the pneumonia of unknown cause in Wuhan is coronavirus | Mask supply in Southeast Asia | Riots in Hong Kong | The situation in Wuhan is grim | Japanese swine fever infection |
| | 3 | Comprehensive prevention and control of infectious diseases | One person has died of pneumonia of unknown cause in Wuhan | Experts interpret pneumonia of unknown cause in Wuhan | Japan has a shortage of masks | Yellow alert for flu in Tianjin | | other |
| | 4 | The government increased access to information | Update on the novel Coronavirus pneumonia in Wuhan | The novel Coronavirus gene sequence is highly similar to SARS | | Wuhan virus laboratory investigated | | |
| | 5 | Prevent coronavirus infection | Notification of confirmed cases | The coronavirus is not same as SARS | | Online teaching by students | | |

| Domain | Num | EP(a) | DR(b) | SRP(c) | CD(d) | EC(f) | ES(h) | PS(i) |
|---|---|---|---|---|---|---|---|---|
| Dt-T3 Topics | 6 | Dou action to defeat the epidemic | Update on COVID-19 cases | Zhong nanshan is certain of human-to-human transmission of COVID-19 | The government has paid for the treatment of all patients in Wuhan | There is a possibility of a wider outbreak of novel Coronavirus | Hats off to the medical staff! Wuhan refueling | Anhua Wu went to the front lines of COVID-19 |
| | 7 | Wearing masks is the key to preventing novel Coronavirus | Sichuan confirmed the first case of wuhan novel pneumonia | China will attend the meeting to share information on the epidemic | Wuhan road | | | |
| | 8 | ECMO technology successfully treated a patient with pneumonia | | Teach you how to wear a mask correctly | | | | |
| | 9 | Folk ancestral special prescription for fighting novel pneumonia | | Traditional Chinese medicine for COVID-19 prevention | | | | |
| | 10 | Henan has banned the sale of live poultry in markets | | | | | | |

### 5.2. Topic Evolution Relationship Mining

After the topic is discovered, this paper mines the evolutionary relationship between topics. KL divergence refers to the information gained or relative entropy used to quantify the asymmetric difference between two probability distributions, P and Q, which can be used to measure the evolution of topic content [32]. As the evolutionary relationship is asymmetric, KL divergence is adopted in this paper to measure the evolutionary relationship. The calculation of KL divergence is shown in Formula (10).

$$\mathrm{KL}(P_{T_m}||P_{T_n}) = \sum\nolimits_{i=1} P^{T_m}(x_i) \times log \frac{P^{T_m}(x_i)}{P^{T_n}(x_i)} \tag{10}$$

where, $P_{T_m}$ represents the distribution of feature words of topic $T_m$, $P^{T_n}$ represents the distribution of feature words of topic $T_n$, $P^{T_m}(x_i)$ represents the probability of feature words $x_i$ in $P_{T_m}$, and $\mathrm{KL}(P_{T_m}||P_{T_n})$ represents the loss of information to approximate with $T_n$ and $T_m$. The smaller the value, the closer $T_n$ is to $T_m$, the more likely $T_n$ is to evolve from $T_m$, and vice versa. At the same time, considering that some of the directly extracted keywords have the same conceptual semantics, but the words are different and therefore cannot be recognized, this study extends the conceptual semantics of the topic feature words based on the CN-Probase conceptual knowledge atlas platform.

The evolution of topics in the source domain and target domain can be divided into four types: one-to-one relationship (OTO), one-to-many relationship (OTM), many-to-one relationship (MTO), and many-to-many relationship (MTM). This paper set the calculation results of KL divergence as the basis to determine whether the evolution between topics takes place. After many tests, we set different KL divergence thresholds to observe the results of the experiment. When KL = 0.25, the classification accuracy of topics with evolutionary relationships was the highest. Therefore, the threshold value of KL divergence is set to 0.25. If the threshold value is lower than 0.25, it indicates that there may be an evolutionary relationship between topics. Taking the scientific research and popularization (SRP) subject as an example, the results of the evolutionary relationship between topics are shown in Table 8.

**Table 8.** The evolutionary relationship of SRP topics.

| Num | Topic Num | Direction | Topic Num | Kullback-Leible (KL) | Evolutionary Relationship |
|---|---|---|---|---|---|
| 1 | c1 | –> | c6 | 0.16 | MTM |
| 2 | c1 | –> | c7 | 0.23 | MTM |
| 3 | c1 | –> | c8 | 0.21 | OTM |
| 4 | c1 | –> | c9 | 0.12 | MTM |
| 5 | c2 | –> | c7 | 0.22 | MTM |
| 6 | c3 | –> | c6 | 0.25 | MTM |
| 7 | c3 | –> | c7 | 0.17 | MTM |
| 8 | c4 | –> | c6 | 0.19 | MTO |
| 9 | c5 | –> | c6 | 0.18 | MTM |
| 10 | c5 | –> | c9 | 0.22 | MTM |

In order to facilitate the construction of the topic knowledge graph, this paper defines four categories of topics, including starting topic, ending topic, single topic, and hot topic. The above method is used to mine the evolutionary relationship of T2 time slice in the source domain and T3 time slice in the target domain, and the evolutionary results are shown in Figure 9.

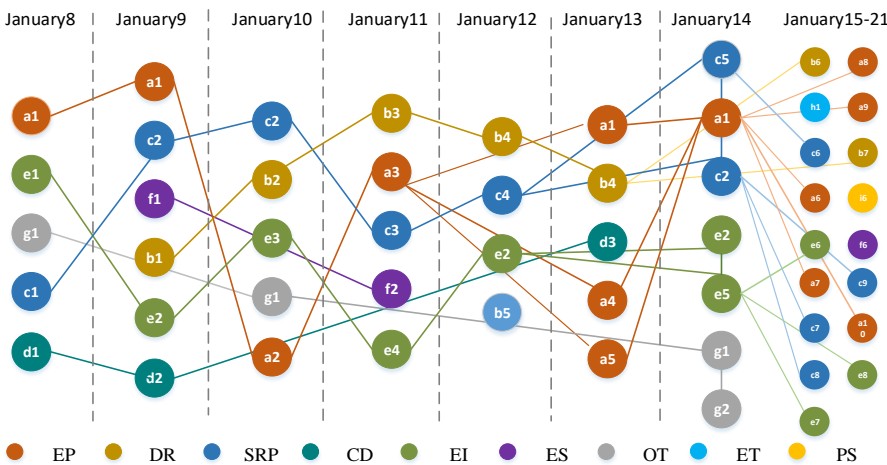

**Figure 9.** Graph of topic evolution over time (8 January 2020–21 January 2021).

### 5.3. Topic Evolution Prediction Based on Knowledge Graph

A knowledge graph is a network with a semantic nature, and its basic constituent unit is the triple of "entity, relation, entity", through which knowledge is expressed in a graph structure [33,34]. Entities are essentially the semantic objects in the real world. Network public opinion entities can be individuals or organizations for opinions through social networks such as media, government, Internet users, social institutions, etc., or incidents causing wide public concern. The entity of this article is extracted from the social network dataset and contains four entity classes: Subject, topic, event, person. Through the research on the defined entities, four relations of <subject, include, topic>, <event, belong to, topic>, <people, belong to, topic> and <people, involve, event> are obtained, as shown in Figure 10, where the ellipse represents the entity category, and the line represents the relation.

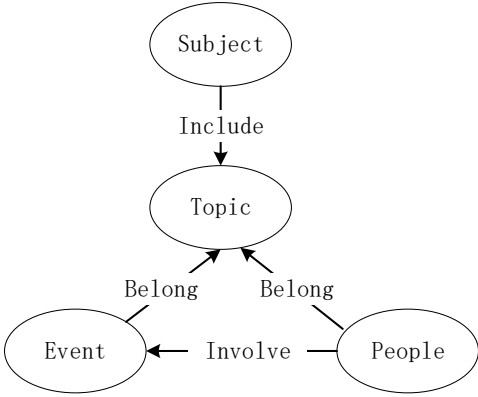

**Figure 10.** Entity, relation model diagram.

The topic knowledge graph was constructed from text data of source and target domains from 25 December 2019 to 7 April 2020, including subjects extracted by LDA model and topics, events, and people extracted by single-pass model. Considering the display effect of the graph, the knowledge graph ignoring the human entities as shown in Figure 11a,b intercepted part of the public opinion topic subgraph with the source domain T2 time slice and the target domain T3 time slice data as the core data, with the human entity added.

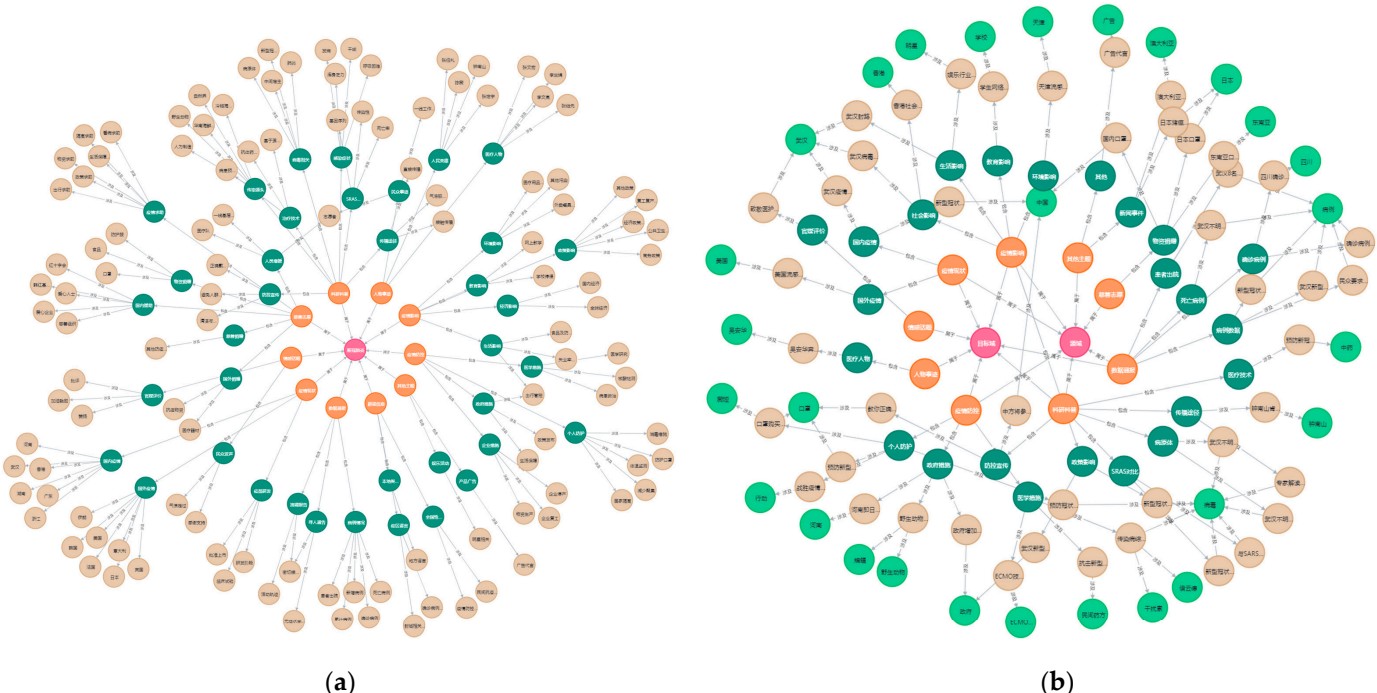

(**a**)  (**b**)

**Figure 11.** This is the topic knowledge graph constructed by using Neo4j graph database: (**a**) is the COVID-19 Topic Knowledge Graph; (**b**) is the topic knowledge graphs of source domain T2 and target domain T3.

This paper takes the topic knowledge graph established by the source domain T2 time slice and the target domain T3 time slice as the research samples and selects five representative online public opinion events in the source domain on 22 January 2020 as test samples. They are, respectively, "Henan has banned the sale of live poultry in the market" (N1), "The Ministry of Transportation launched the level II emergency response" (N2), "Wuhan set up the epidemic prevention and control headquarters" (N3), "CDC will announce the first novel Coronavirus case in the US" (N4), and "Strict prevention of Novel Coronavirus in Spring Festival Transportation" (N5).

By traversing the nodes in the topic knowledge graph, cosine similarity was used to calculate the similarity between the key words of test events N1–N5 and the key words of nodes in the knowledge graph, and the node with the highest similarity was found. However, the accuracy of node prediction results in the knowledge graph of the target domain is low, so nodes with an evolutionary relationship between the source domain and the target domain are transferred to improve the accuracy of topic prediction in the target domain so as to realize topic prediction in the target domain.

According to the similarity calculation of topic nodes in knowledge graph, we get the source domain N1–N5 test events subject prediction results, as shown in Table 9. At the same time, we traverse the nodes to find the target domain generalization topics, as shown in Table 10. This paper extracts the weight of the biggest prediction of top 10 subjects and topics, and the accuracy of the prediction results was judged manually. The correct prediction was recorded as Y, and the wrong prediction was recorded as N.

Additionally, this paper adopts the MRR (Mean Reciprocal Rank) indicator to measure the topic prediction effect. The prediction accuracy of N1–N5 test events calculated by MRR method is shown in Table 11, and the mean value is taken as the final result, according to which the subject prediction accuracy is 82.6% and the topic prediction accuracy is 70.46%, indicating that the model has good prediction effect and strong applicability.

**Table 9.** N1–N5 Test event target domain subject division prediction results.

| Num | N1 | Sim | Y/N | N2 | Sim | Y/N | N3 | Sim | Y/N | N4 | Sim | Y/N | N5 | Sim | Y/N |
|---|---|---|---|---|---|---|---|---|---|---|---|---|---|---|---|
| 1 | RC | 0.396 | N | EP | 0.305 | Y | EC | 0.498 | Y | EC | 0.331 | Y | SRP | 0.408 | Y |
| 2 | EI | 0.377 | Y | EI | 0.225 | Y | SRP | 0.461 | N | EP | 0.241 | N | EP | 0.379 | Y |
| 3 | EP | 0.370 | Y | EC | 0.215 | Y | DR | 0.457 | N | SRP | 0.240 | N | PS | 0.361 | N |
| 4 | EC | 0.363 | Y | ES | 0.210 | N | RC | 0.411 | Y | CD | 0.203 | N | CD | 0.358 | N |
| 5 | DR | 0.358 | N | SRP | 0.196 | N | EP | 0.397 | Y | DR | 0.201 | Y | ES | 0.352 | N |
| 6 | CD | 0.346 | N | CD | 0.191 | N | ES | 0.391 | N | RC | 0.195 | N | EI | 0.340 | N |
| 7 | SRP | 0.344 | N | DR | 0.181 | N | CD | 0.368 | N | ES | 0.193 | N | EC | 0.338 | N |
| 8 | PS | 0.342 | N | PS | 0.179 | N | EI | 0.340 | N | EI | 0.192 | N | RC | 0.330 | N |
| 9 | ES | 0.327 | N | RC | 0.172 | N | PS | 0.310 | N | PS | 0.165 | N | DR | 0.314 | N |

**Table 10.** N1–N5 test the target domain generalization topic prediction results of events.

| Num | N1 | Sim | Y/N | N2 | Sim | Y/N | N3 | Sim | Y/N | N4 | Sim | Y/N | N5 | Sim | Y/N |
|---|---|---|---|---|---|---|---|---|---|---|---|---|---|---|---|
| 1 | Virus associated | 0.312 | Y | Government measures | 0.489 | Y | Domestic outbreak | 0.616 | Y | Government measures | 0.374 | Y | Prevention publicity | 0.506 | Y |
| 2 | Public opinion | 0.332 | Y | Life influence | 0.356 | Y | Virus associated | 0.535 | Y | Foreign outbreak | 0.341 | Y | Virus associated | 0.430 | Y |
| 3 | Public events | 0.359 | N | Education influence | 0.261 | Y | Infection Source | 0.531 | N | Virus associated | 0.330 | Y | Personal protective | 0.422 | Y |
| 4 | Prevention publicity | 0.338 | Y | Virus associated | 0.246 | Y | Local rumors | 0.521 | Y | Domestic outbreak | 0.320 | Y | Infection Source | 0.398 | N |
| 5 | Infection Source | 0.339 | Y | Domestic outbreak | 0.244 | Y | confirmed case | 0.516 | Y | Infection Source | 0.306 | Y | Government measures | 0.390 | Y |
| 6 | Domestic outbreak | 0.304 | Y | Medical measures | 0.226 | N | Prevention publicity | 0.484 | Y | confirmed case | 0.283 | Y | Life influence | 0.387 | Y |
| 7 | Medical measures | 0.346 | N | Public opinion | 0.225 | Y | Medical measures | 0.427 | Y | Prevention publicity | 0.256 | Y | Material donations | 0.387 | N |
| 8 | Environmental impact | 0.452 | N | Public events | 0.223 | N | dead csae | 0.447 | N | Life influence | 0.250 | N | Treatment technology | 0.377 | N |
| 9 | Material donations | 0.313 | N | Environmental impact | 0.215 | N | Environmental impact | 0.421 | N | Education influence | 0.243 | N | Public opinion | 0.318 | Y |
| 10 | Education influence | 0.303 | N | Material donations | 0.185 | N | Material donations | 0.321 | N | Foreign donors | 0.229 | N | Environmental impact | 0.303 | N |

**Table 11.** Results Prediction Accuracy Test.

| Test Events | Subject Prediction Accuracy | Topic Prediction Accuracy |
|---|---|---|
| N1 | 77.80% | 74.00% |
| N2 | 88.90% | 73.30% |
| N3 | 74.10% | 68.30% |
| N4 | 77.80% | 70.00% |
| N5 | 94.40% | 67.00% |
| Average | 82.60% | 70.46% |

## 6. Conclusions and Future Work

We implemented cross-domain transfer learning between two social network platforms and propose an improved TC-LDA model to measure the similarity between the two domains. Compared with the LDA model, the TC-LDA model has improved *Accuracy*, *Procision*, *Recall* and *F − Measure* values. Based on the results of transfer learning, we built a topic knowledge graph by using the Neo4j graphics database and conducted experiments to predict the evolution of popular topics in new emergencies. Experimental results show that knowledge graph technology is effective in popular topic prediction.

In this paper, entities, relationships, and attributes were extracted from COVID-19 emergency information to construct knowledge maps and predict topics. On the one hand, the effect of online public opinion analysis could be improved, so that government departments could make predictions in advance and take corresponding guidance measures in a timely manner to prevent public opinion from blindly expanding. On the other hand, the construction of the topic knowledge map based on the characteristics of the COVID-19 event also expands the application of the knowledge map in the field of public opinion

analysis of emergencies on social networks, which can provide theoretical reference for studies in similar fields.

In future research, the graph convolutional network will be considered to model richer sentence semantics. In future research, multiple source domain data will be considered to be used for transfer learning to improve the prediction accuracy of information popular topics.

**Author Contributions:** Data curation, X.C.; methodology, Q.Q. and S.C.; project administration, X.C.; writing—original draft, X.C.; writing—review & editing, C.W. All authors have read and agreed to the published version of the manuscript.

**Funding:** This research was funded by the National Natural Science Foundation of China (No. 71801047); Beijing Philosophy and Social Sciences Planning Project (No. 17GLC052); Fundamental Research Funds for the Central Universities in UIBE (CXTD12-04); The Foundation of Key Laboratory of Trustworthy Distributed Computing and Service (BUPT), Ministry of Education; UIBE Distinguished Young Scholars Project of UIBE.

**Institutional Review Board Statement:** Not applicable.

**Informed Consent Statement:** Not applicable.

**Data Availability Statement:** Not applicable, the study does not report any data.

**Conflicts of Interest:** The authors declare no conflict of interest.

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
