# Peer review of "Cross-Domain Transfer Learning Prediction of COVID-19 Popular Topics Based on Knowledge Graph"

_futureinternet, doi:10.3390/fi14040103_

Round 1

Reviewer 1 Report

* lacks a lot of reference pointers (a few examples: DSN, "Baidu deactivation Thesaurus", "Harbin Institute of technology deactivation Thesaurus" and "Sichuan University Machine Intelligence Laboratory deactivation Thesaurus")

* fig.2 needs to be explained in detail: what exactly is transferred and how? it gives the impression that the authors only mix data in D_s and some data in D_t as training data. a lot of notions are not explained (F, h)

* the terms in fig.6 and some tables are not translated in english. non-chinese speakers will find it hard to understand

* what is the fraction of training/test data in the experiments?

* section 5 "Prediction of Popular Topics Based on Knowledge Graph". how is it related to transfer learning in the previous sections?

* how to objectively evaluate the result in Table 6 and 8? the accuracy of the prediction results was judged manually.

* how to make sure the method is applicable on big data if some steps are done "manually" (the authors mentioned it a couple of times)?

Reviewer 2 Report

Why KL divergence is used to evaluate the evolutionary relationships, not much of details are present in the paper. Could you please elaborate on that in detail?

Why KL of 0.25 is taken as a threshold, is it data-driven or any scientific choice?

The plot (Figure 9) is not legible to read, could you please add a high-quality plot?

The tables (Table 8, 9) are not clean as well, could you please add a high-quality table. 

The paper is not very novel in the context of the methodology, however, novel in the context of the data domain. The paper is OK for scientific merit, and English is good. The paper is OK to publish with minor revisions such as the figures/tables.  The image quality is not good, so only this thing to be corrected.

Round 2

Reviewer 1 Report

The authors addressed the points in the review and I agree to accept the revised manuscript.

Author Response

Dear reviewer:

Thank you very much for your positive and constructive comments and suggestions on our manuscript entitled “Cross-Domain Transfer Learning Prediction of COVID-19 Popular Topics Based on Knowledge Graph”.  We hope that the correction will meet with approval. Once again, thank you very much for your comments and suggestions。

Thank you and best regards.

Reviewer 2 Report

Figure 6 (the lower figure) does not have good quality when we zoom the page, could you please make the image good?

The image quality of Table 1 is not good when we zoom the page, could you please make the image good? Same for Table 7.

Author Response

Dear reviewer:

Thank you very much for your positive and constructive comments and suggestions on our manuscript entitled “Cross-Domain Transfer Learning Prediction of COVID-19 Popular Topics Based on Knowledge Graph” (Manuscript ID:futureinternet-1626184) .We hope that the correction will meet with approval.

Replies to the reviewer’s comments:

1.Figure 6 (the lower figure) does not have good quality when we zoom the page, could you please make the image good?

Response:Done, as requested. As suggested by the reviewer, the figure 6 in the article have been replaced with a high-quality picture. (Added to page 7)

2.The image quality of Table 1 is not good when we zoom the page, could you please make the image good? Same for Table 7.

Response:Done, as requested. As suggested by the reviewer, we have tried our best to add the high-quality Table1,Table7 and Table10. (Added to page 8,page12 and page15)

Once again, thank you very much for your comments and suggestions。

Thank you and best regards.

Yours sincerely.

Name: Qixing Qu

E-mail: qqxing@uibe.edu.cn